# Spatial and temporal scales of variability for indoor air constituents

Pascale S. J. Lakey [1,14], Youngbo Won[2,14], David Shaw [3], Freja F. Østerstrøm [3], James Mattila [4], Emily Reidy[5], Brandon Bottorff[5], Colleen Rosales [5], Chen Wang [6,13], Laura Ampollini[7], Shan Zhou [8,9], Atila Novoselac[10], Tara F. Kahan[8,11], Peter F. DeCarlo [12], Jonathan P. D. Abbatt [6], Philip S. Stevens[5], Delphine K. Farmer[4], Nicola Carslaw[3], Donghyun Rim[2✉] & Manabu Shiraiwa[1✉]

Historically air constituents have been assumed to be well mixed in indoor environments, with single point measurements and box modeling representing a room or a house. Here we demonstrate that this fundamental assumption needs to be revisited through advanced model simulations and extensive measurements of bleach cleaning. We show that inorganic chlorinated products, such as hypochlorous acid and chloramines generated via multiphase reactions, exhibit spatial and vertical concentration gradients in a room, with short-lived ·OH radicals confined to sunlit zones, close to windows. Spatial and temporal scales of indoor constituents are modulated by rates of chemical reactions, surface interactions and building ventilation, providing critical insights for better assessments of human exposure to hazardous pollutants, as well as the transport of indoor chemicals outdoors.

[1] Department of Chemistry, University of California, Irvine, Irvine, CA, USA. [2] Department of Architectural Engineering, Pennsylvania State University, University Park, PA, USA. [3] Department of Environment and Geography, University of York, York, UK. [4] Department of Chemistry, Colorado State University, Fort Collins, CO, USA. [5] Department of Chemistry and O'Neill School of Public and Environmental Affairs, Indiana University Bloomington, Bloomington, IN, USA. [6] Department of Chemistry, University of Toronto, Toronto, Canada. [7] Department of Civil, Architectural, and Environmental Engineering, Drexel University, Philadelphia, PA, USA. [8] Department of Chemistry, Syracuse University, Syracuse, NY, USA. [9] Department of Civil and Environmental Engineering, Rice University, Houston, TX, USA. [10] Department of Civil, Architectural, and Environmental Engineering, University of Texas, Austin, TX, USA. [11] Department of Chemistry, University of Saskatchewan, Saskatoon, Canada. [12] Department of Environmental Health and Engineering, Johns Hopkins University, Baltimore, MD, USA. [13] Present address: School of Environmental Science and Engineering, Southern University of Science and Technology, Shenzhen 518055, China. [14] These authors contributed equally: Pascale S. J. Lakey, Youngbo Won. ✉email: dxr51@psu.edu; m.shiraiwa@uci.edu

People spend on average 90% of their time indoors and even longer especially in their homes during the current COVID-19 pandemic. Concentrations of indoor gaseous compounds and aerosol particles are often much higher compared to outdoors owing to indoor emission sources including human activities such as cleaning and cooking[1,2]. Hypochlorite bleach is an effective disinfectant that kills a wide variety of microorganisms and is increasingly used to control infectious disease spread in various indoor locations including schools, hospitals, and residential buildings[3,4]. Following bleach use, a number of chlorinated compounds including hypochlorous acid (HOCl) and molecular chlorine gas (Cl$_2$) can be released[5,6], which are hazardous by causing skin lipid oxidation[7] and cytotoxic injury in the respiratory tract[4]. The House Observations of Microbial and Environmental Chemistry (HOMEChem) campaign[8] has revealed that a series of multiphase reactions involving nitrite (NO$_2^-$) and ammonia (NH$_3$) in the applied bleach onto a floor can lead to the formation of nitryl chloride (ClNO$_2$) and chloramines (e.g., NCl$_3$)[5], which have strong irritation effects with the potential to damage tissues[4]. Bleach cleaning chemistry also produces several toxic compounds including isocyanates, cyanogen chloride, and chlorocarbons[9]. As such, bleach can pose an increased risk of respiratory infections and symptoms such as wheezing and asthma[3,10].

Historically indoor air constituents have been assumed to become well mixed and homogeneously distributed after being introduced into ventilated indoor environments[11]. Hence, indoor measurements are mostly conducted at a single location in a room and at a fixed height and there have been only a few measurements of spatial and vertical distributions of gas pollutants and particulate matter[12,13]. While computational fluid dynamics (CFD) simulations have been applied to resolve indoor air flows and spatial distributions of non-reactive indoor species[14], indoor chemistry models often employ a box model with the concept of deposition velocity assuming that there is a well-mixed core region separated from indoor surfaces by boundary layers[11]. However, this assumption may not be warranted for reactive and short-lived species such as radicals and bleach cleaning products. To better quantify human exposure to indoor pollutants, it is essential to evaluate spatial distributions and temporal scales of emitted compounds, which are currently poorly understood.

To elucidate the spatial and temporal scales of variability of indoor air pollutants, in this paper we go far beyond earlier studies by integrating multiple indoor models including gas-phase chemistry modeling, multiphase kinetic modeling, and computational fluid dynamics (CFD) simulations[15] to simulate extensive measurements of a bleach cleaning event from HOMEChem. We show that ·OH radicals and bleach cleaning products exhibit spatial and vertical concentration gradients in a room as modulated by rates of chemical reactions, surface interactions, and ventilation.

## Results

**Integrated modeling for HOMEChem measurements**. We developed a multiphase kinetic model to treat formation and loss of bleach products to simulate gas-phase measurements performed during HOMEChem (Fig. 1a)[5]. It treats outdoor–indoor air exchange, gas-phase reactions, photolysis, wall loss, heterogeneous reactions at indoor surfaces and particles, and aqueous reactions in the aqueous bleach, while assuming that species would be mixed homogeneously in the room where the bleach was applied[5] (see Supplementary Methods). Transport of semi-volatile species between the gas phase and the bleach requires transport through a boundary layer adjacent to the bleach surface,

which is resolved explicitly in the model[16]. In addition, a detailed photochemical box model, the INdoor Detailed Chemical Model (INDCM) with the Master Chemical Mechanism, was used to quantify the radical production rates and refine the predicted radical concentrations[17] (see Supplementary Methods). Most measurements were conducted at one location (P2) in the kitchen, while ·OH was measured in the sunlit zone next to the window at P7 (see Fig. 1b and Supplementary Fig. 1).

As shown in Fig. 1c, the results from these well-mixed models can be directly compared with the measurements. The INDCM model successfully captures ·OH radical concentrations and the multiphase kinetic model reproduces the measured temporal variation of HOCl, NCl$_3$, and NH$_3$. These results, however, do not reflect the potential heterogeneous distribution of reactive species in an indoor space; a CFD model is necessary to resolve this. While the gas-phase chemistry model and multiphase kinetic model treat comprehensive and detailed chemistry, it is computationally too expensive and unfeasible to treat all of these gas and multiphase reactions in the CFD. To circumvent this hurdle, we constrained the CFD with key inputs from the detailed models: the INDCM provided production rates and reactivity of ·OH radicals, while the multiphase kinetic model provided HOCl, ClNO$_2$, NCl$_3$, and NH$_3$ concentrations right above the bleach surface over time as controlled by aqueous reactions in the bleach. These models also identified critical gas-phase reactions as well as specific photolysis rates, rate coefficients, and uptake coefficients to surfaces to be included in the CFD (Supplementary Methods and Supplementary Table 1).

By resolving spatial heterogeneity, the CFD model reproduces the dynamic concentration changes at the sampling points remarkably well (Fig. 1c). After the bleach containing NaOCl is applied to the floor for 10 min, HOCl is formed in the aqueous bleach and volatilized to the gas phase. HOCl undergoes heterogeneous reactions on acidic particles or indoor surfaces, leading to the formation of Cl$_2$. In the aqueous bleach, HOCl reacts with nitrite (NO$_2^-$) that is largely present on indoor surfaces as a reservoir of HONO[18], to generate ClNO$_2$ which can partition into the gas phase. NH$_3$, emitted by human occupants and off-gassing from building materials and indoor surfaces[19], partitions into the aqueous bleach to participate in a series of reactions with HOCl to generate NCl$_3$[5], leading to an increase of NCl$_3$ and a decrease of NH$_3$ in the gas phase. Afterwards, the bleach products decayed faster than the air exchange rate, which is also captured very well by accounting for deposition to indoor surfaces.

Model simulations reveal that the observed enhancement of ·OH radicals during the bleach cleaning event can be mainly explained by a cascade of reactions initiated via Cl$_2$ photolysis: the formed Cl radicals react with volatile organic compounds (VOCs) to generate peroxy and alkoxy radicals, which propagate to HO$_2$ and then ·OH through reactions involving NO. Gas-phase model simulations indicate that this process accounts for >90% of ·OH production, while ·OH radicals can also be generated via photolysis of HOCl and HONO[5]. The generated ·OH radicals react rapidly with a number of indoor gas-phase species including NO$_x$ and VOCs with an estimated ·OH reactivity of 65 s$^{-1}$ during the cleaning event (see Supplementary Methods). The remarkable level of agreement between measurements and simulations for radicals and reaction products has been made possible by effectively resolving complex physical and chemical processes as well as indoor air flow and spatial heterogeneity.

Horizontal and vertical distributions in Fig. 2 show that high concentrations of ·OH radicals are confined only to the solar radiation zone where they are generated via photolysis, while their concentration is low in the dark zone due to depletion through loss reactions. However, the products of ·OH radical

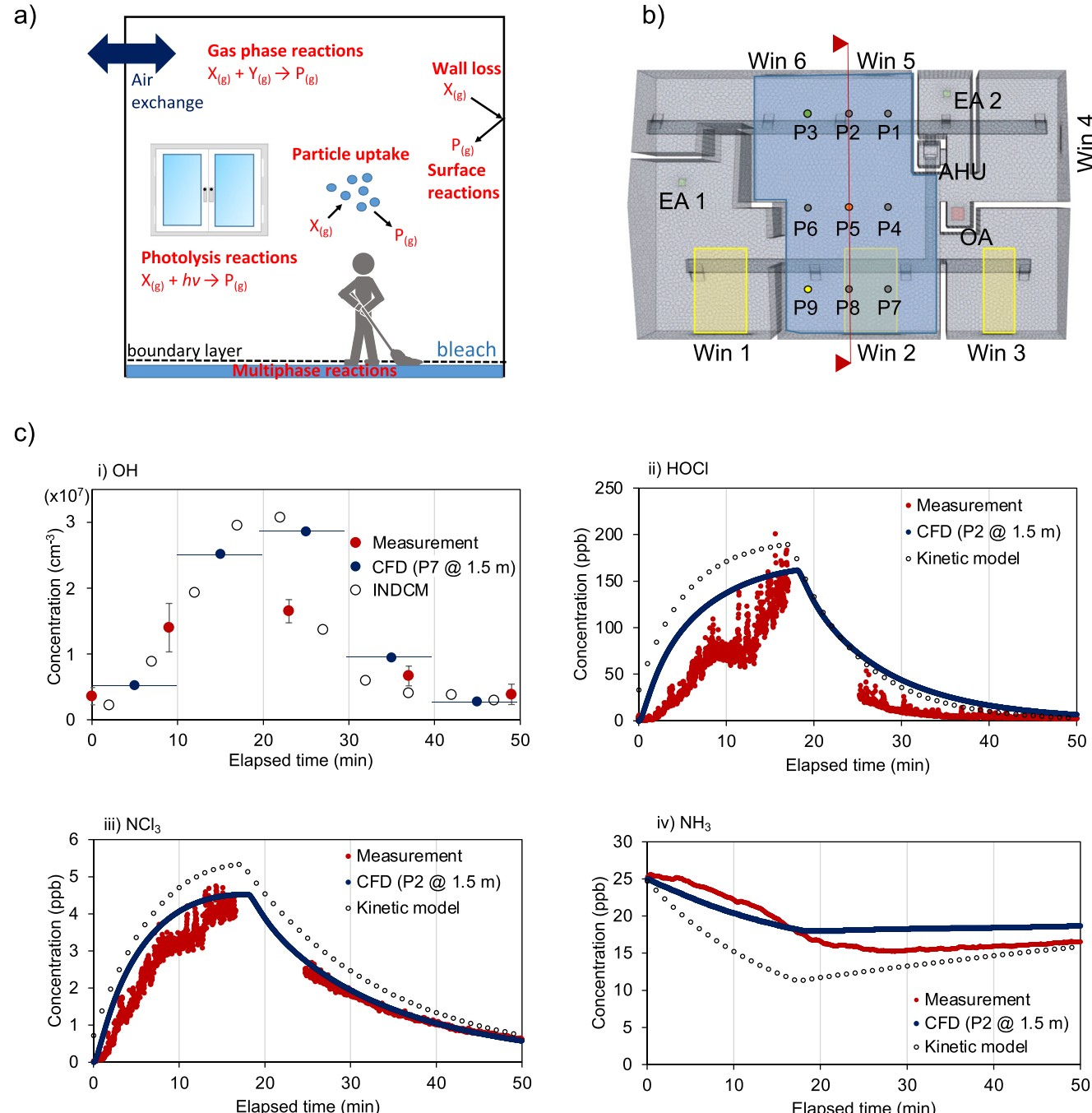

**Fig. 1 Integrated modeling of bleach cleaning events. a** A schematic of the kinetic model to simulate a cleaning event at the HOMEChem campaign. **b** The floor plan of the test house and computational fluid dynamics modeling geometry (Win: window, AHU: air handling unit, EA: exhaust air, OA: outside air). The yellow marks are solar radiation zones and blue marks are cleaning area. Nine points (P1–P9) at 1.5 m above the cleaning floor surface are the calculation points in CFD simulations. The vertical red line represents the cross-section used for the vertical maps presented in Fig. 2. **c** Temporal evolution of (i) OH, (ii) HOCl, (iii) NCl$_3$, and (iv) NH$_3$ as measured (red) and simulated by the CFD (dark blue), the INDCM (open markers in **c** (i)), and the multiphase kinetic model (open markers in **c** (ii–iv)). The error bars in (**c**) represent the 1σ precision of the OH measurements and are separate from the calibration accuracy (±18%, 1σ).

reactions such as HCHO and OVOCs in these zones will have longer lifetimes, thus increasing the effective spatial impact of ·OH radical production. HOCl, ClNO$_2$, and NCl$_3$ are emitted from the cleaning surface, resulting in vertical concentration gradients and higher concentrations in the living room compared to other rooms. Note that although the air handling unit circulates a fairly large amount of indoor air in the whole house at a mixing rate of 8 h$^{-1}$ with all room doors open, the cleaning

products are primarily concentrated in the living room. Even in the living room, these products are confined to the area near the corner (P1) because of the non-uniform indoor airflow, showing 30–50% higher concentrations than at other points. Cl$_2$ also exhibits similar spatial distributions (Supplementary Fig. 2), reflecting that Cl$_2$ is mainly produced where HOCl is more concentrated. NH$_3$ is relatively homogeneously distributed, with a few ppb lower mixing ratio in the cleaning area compared to

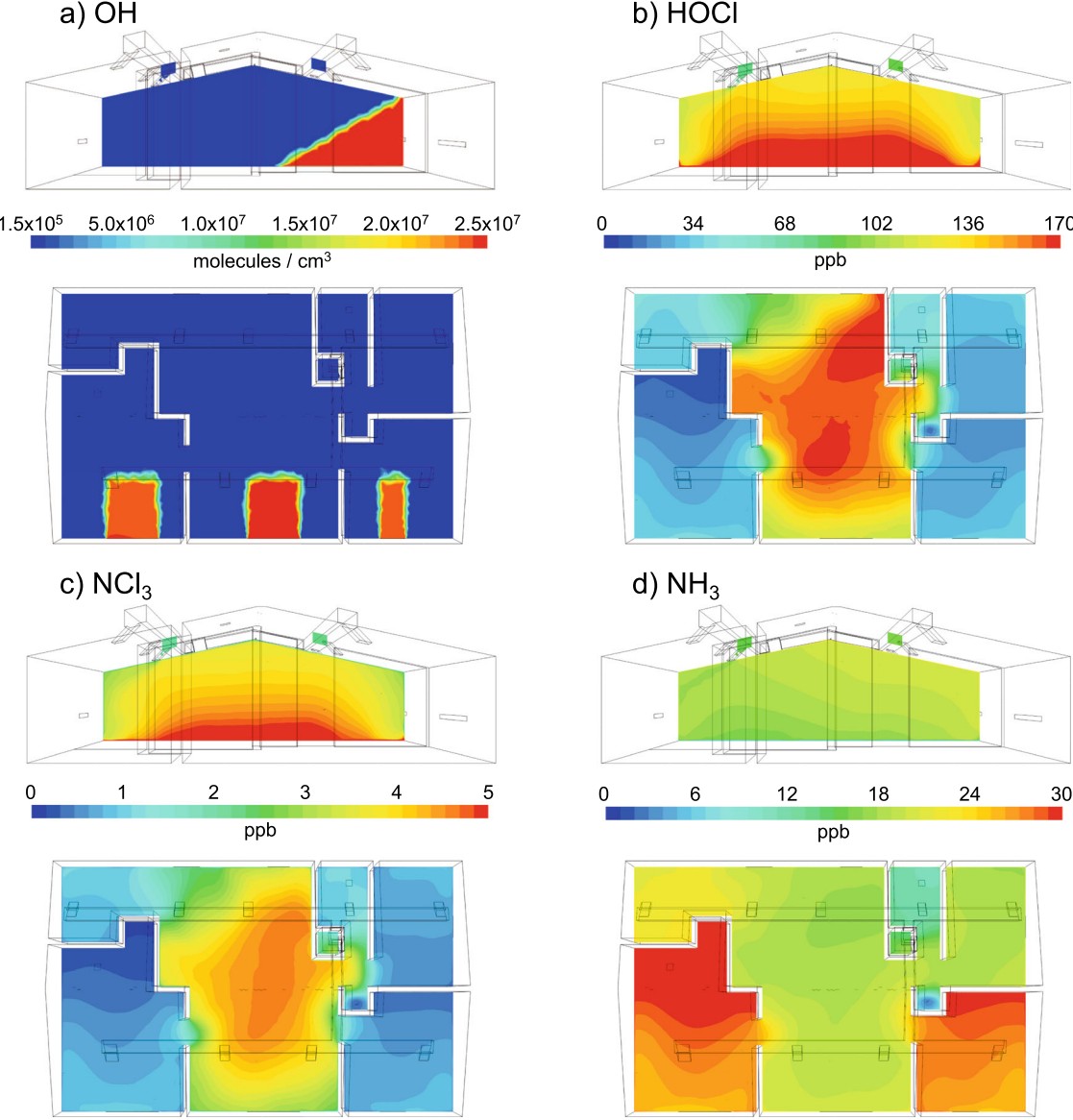

**Fig. 2 Spatial distributions of bleach products.** Horizontal and vertical spatial distributions of (**a**) OH, (**b**) HOCl, (**c**) NCl$_3$, and (**d**) NH$_3$ at 18 min after the beginning of the cleaning. Horizontal maps represent 1.5 m above the floor and vertical maps represent sections with the red line in Fig. 1b.

other rooms due to uptake into the bleach followed by aqueous reactions.

**Spatial and temporal variations of indoor species.** Similar to the atmosphere[20], indoor air can be regarded as a highly dynamic chemical reactor. A variety of chemical species is introduced and removed over a wide range of spatial and temporal scales, depending on rates of ventilation, photolysis, chemical reactions, and deposition as well as room and building sizes[1,2]. We estimate half-lives of representative indoor chemical species by considering a typical air exchange rate of 0.5 h$^{-1}$ and reaction rates with typical indoor concentration levels of ·OH (3 × 10$^5$ cm$^{-3}$), O$_3$ (4 ppb), NO$_x$ (7 ppb), and VOCs (100 ppb) as well as typical photolysis and surface deposition rates (see Supplementary Table 2). Then, spatial scales or the average distance traveled can be estimated by considering a typical indoor air flow velocity of 0.03 m s$^{-1}$, corresponding to an air exchange rate[21] of 0.5 h$^{-1}$. The results of this analysis are depicted in Fig. 3, in which three distinct scales emerged:

1. Microscale: Processes occurring on the spatial scale of <~0.1 m and affecting phenomena only in proximity to emission sources or locations where compounds are generated in tiny eddies of a centimeter or less. Near the emission sources, short-lived radical species (with lifetimes up to ~10 s) such as Cl, NO$_3$, and RO$_x$ (=·OH + HO$_2$· + RO$_2$·) exhibit sharp spatial gradients and their temporal scales are determined mainly by reaction rates, and only marginally affected by deposition and ventilation rates (see Supplementary Methods).

2. Room scale: Processes that exceed the microscale but still occur within a room (~0.1–10 m). Moderately long-lived species (with time scales of ~10 s–10 min) such as NH$_3$, NO, Cl$_2$, and O$_3$ would exhibit spatial gradients within a room. The temporal and spatial distributions of these species are controlled by both chemical processes and indoor air flow conditions. For NH$_3$ and semi-volatile organic compounds (SVOCs) that may undergo reversible partitioning to indoor surface reservoirs[18], the true spatial

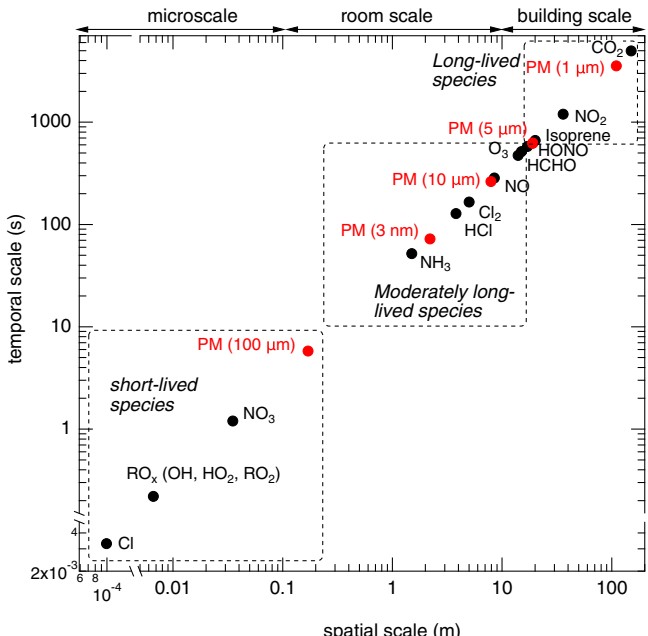

**Fig. 3 Spatial and temporal scales of variability for indoor species.** Spatial and temporal scales of gas-phase species and particulate matter with different particle diameters indoors with an air exchange rate of 0.5 h$^{-1}$.

gradients are likely reduced from the model predictions due to surface emissions.

3. Building scale: Phenomena occurring in a plume on scales larger than a room (>~10 m), possibly affecting other rooms and the entire building by circulation and even being transported outdoors. Long-lived species such as VOCs, NO$_2$, and CO$_2$ are mostly well mixed within the indoor space. Their temporal scales are mainly controlled by ventilation rates. During HOMEChem, HONO was measured in two different locations (P2 and P7), showing very similar concentrations (see Supplementary Fig. 4)[22].

A better understanding of spatial distributions of indoor species is highly critical for accurate assessments of human exposure to indoor oxidants and SVOCs including toxic chlorinated and nitrogenated VOCs[9,23]. The widely applied concept of deposition velocity, which expresses the species flux density to the surface divided by its concentration in the uniformly mixed core region, may need to be revisited[11] for simulating short-lived and moderately long-lived species. Note that this analysis of temporal and spatial scales implicitly assumes that spatial gradients are driven by a perturbation such as cleaning, cooking, and other activities at steady-state conditions; there will not be the same gradients in the absence of a perturbation. Spatial heterogeneity in photon fluxes also leads to spatial gradients of photoactive species. Resolving mass transport and chemical reactions in the boundary layer[16] and on indoor surfaces would be required for an accurate description of deposition processes[24].

The spatial scale indoors is several orders of magnitude smaller than for species in the ambient atmosphere[20]. Because of relatively low air exchange rates in residences, non-reactive gas-phase species remain indoors for 3–4 h. At higher air exchange rates that are often deployed in industrial buildings with mechanical ventilation, the temporal and spatial scales of moderately long-lived and long-lived species would both decrease as the species are transported to the ambient atmosphere at a faster rate (see Supplementary Fig. 3). A recent study has found that the use of volatile chemical products (VCPs, including

pesticides, coatings, adhesives, cleaning agents, and personal care products) constitutes half of fossil-fuel VOC emissions in industrialized cities[25]. VCPs are mostly emitted indoors; however, they are transported outdoors, significantly affecting air quality through the formation of ozone and secondary organic aerosols[25,26]. The analysis in Fig. 3 implies that SVOCs may also be generated indoors and emitted to the ambient atmosphere depending on their reactivity and the ventilation conditions.

Figure 3 also includes spatial and temporal variations of particulate matter (PM) with different particle diameters of 3 nm, 10 nm, 1 μm, 10 μm, and 100 μm, which determine the particle deposition velocity and residence time in indoor environments[14,27] (see Supplementary Table 2). For larger particles with a diameter of 100 μm, they settle to the floor in less than few seconds and within 1 m, mainly due to gravitational settling. Ultrafine particles (1–10 nm) are also relatively short-lived because of particle losses via Brownian and turbulent diffusion. Due to their high mobility, they readily stick to indoor surfaces or are scavenged by bigger particles. On the other hand, 1–10 μm particles are much more persistent in indoor environments with average residence times exceeding minutes and up to 1 h. Their residence times are comparable to the time scale of ventilation rates, so these particles can be transported to other people's breathing zone in indoor environments and can play a critical role as an airborne carrier of infectious pathogens such as SARS-COV2[28,29] as well as for exposure to thirdhand smoke species that have partitioned into indoor particles[30].

## Conclusions

In summary, we demonstrate that heterogeneous distributions of indoor air pollutants can exist for short-lived and moderately long-lived compounds, in contrast to the traditional assumption of homogeneous mixing. The spatial and temporal scales are controlled by gas-phase and multiphase reactions, deposition as well as indoor air flow and outdoor–indoor air exchange. Among these factors, surface interactions may be least characterized and quantified, despite their importance becoming increasingly clear[24,31]. Different surface and environmental conditions including temperature, humidity, light, and surface pH would be critical for heterogeneous reactions at indoor surfaces[24,31] as well as surface stability of SARS-COV2[32]. In addition, the presence of organic films on indoor surfaces can impact thermodynamics and kinetics of SVOC partitioning[33,34]. Further elucidation of these aspects will improve assessments on indoor air quality, human exposure to indoor pollutants, and indoor–outdoor transport of chemical compounds.

## Methods

**HOMEChem campaign.** The House Observations of Microbial and Environmental Chemistry (HOMEChem) campaign and the bleach experiments that occurred during the campaign have previously been described in detail[5,35]. The campaign took place in a 111 m$^2$ 3-bedroom, 2-bathroom test house in Austin, Texas in June 2018. In this work we focus only on the bleaching experiment that occurred on the 8th June as part of a 'layered' experiment, meaning that cooking had happened prior to bleaching. A bleach solution was applied to the kitchen and living room floor, which had a combined surface area of 40 m$^2$, at 17:35. Measurements of gas-phase species concentrations were made prior to and during the experiment using a variety of instruments including a time-of-flight chemical ionization mass spectrometer (TOF-CIMS), a cavity ring-down spectrometer and a laser-induced fluorescence using the fluorescence assay with gas expansion technique instrument (LIF-FAGE). Two separate TOF-CIMS instruments were deployed to sample air from the kitchen (P2): one with utilizing iodide chemical ionization to measure HOCl, Cl$_2$, ClNO$_2$, and NCl$_3$ and another utilizing acetate chemical ionization to measure HONO. The cavity ring-down measured NH$_3$ sampled in the kitchen, while the LIF-FAGE instrument measured ·OH, HO$_2$·, and HONO next to the living room window. Spectrally resolved solar irradiance was measured with a hand-held spectrometer collocated with the LIF-FAGE instrument. The air exchange rate during the period of the bleaching experiment on the 8th June was controlled by a heating ventilation and air conditioning (HVAC) system and was measured as ~0.7 h$^{-1}$.

**Modeling**. The multiphase kinetic model treats various processes including air exchange, uptake to particulate matter and indoor surfaces, photolysis, gas-phase reactions and reactions in the aqueous bleach, and transport of semi-volatile species through a boundary layer above the floor (Fig. 1a and Supplementary Methods). The kinetic model provided inputs to the CFD model including the concentrations of HOCl, ClNO$_2$, chloramines, and NH$_3$ directly above the bleach surface at different times. The CFD model geometry was designed by mimicking air flow and emission conditions of the bleach products observed in the measurement campaign. The CFD model resolves a total of 11 chemical reactions (Supplementary Table 1), solar radiation through windows, surface uptake, and the turbulent indoor air flow (Supplementary Methods). Modeling of ·OH concentrations was carried out using the INDCM (INdoor Detailed Chemical Model)[17], a near-explicit photochemical box model constructed based on a comprehensive chemical mechanism. INDCM also treats exchange with outdoors, internal emissions, photolysis, and deposition to surfaces (Supplementary Methods).

## Data availability

The HOMEChem data is available at the OSF webpage https://osf.io/ykj27/.

## Code availability

The codes used to generate the data in the current study are available from the corresponding author on reasonable request.

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

## Acknowledgements

We acknowledge Dr. Paula Olsiewski and Dr. Marina Vance for their leadership and the entire HOMEChem Science Team for conducting the experiment. This work was funded by the Alfred P. Sloan Foundation (G-2020-13912; G-2019-12306; G-2019-12442; G-2017-9944; G-2019-11404; G-2018-11062; G-2018-10083; G-2017-9796).

## Author contributions

M.S. and D.R. designed and oversaw the study. P.L. and M.S. conducted multiphase kinetic modeling. Y.W. and D.R. conducted CFD simulations. D.S., F.Ø., and N.C. conducted INDCM modeling. E.R., C.R., and P.S. conducted MCM modeling. J.M., E.R., B.B., C.R., C.W., L.A., S.Z., A.N., T.K., P.D., J.A., P.S., and D.F. conducted HOMEChem measurements. P.L., Y.W., D.R., and M.S. wrote the manuscript with inputs from all coauthors. All coauthors discussed the results and contributed to manuscript editing.

## Competing interests

The authors declare no competing interests.
