## [Peer Review File. · Communications Chemistry]

Reviewers' comments:

Reviewer #1 (Remarks to the Author):

In this work, the authors demonstrated that spatial and temporal scales of indoor constituents including gas-phase oxidants, gas-phase species and PM can be modulated by rates of chemical reactions, surface interactions and building ventilation during the bleaching experiment in the HOMEChem campaign in a combination of measurements and CFD simulations. The findings of this work provides greater insights for better assessing the human exposure to gas-phase species and PM in indoor environments. The paper is very well written and concise. I support the publication of this work with minor comments.

Comments

Line 71, "Most measurements were conducted at one location (P2) in the kitchen, while OH was measured in the sunlit zone next to the window at P7 (see Fig. 1b)." Why P2 in the kitchen was chosen for the most of the measurements? What were the results measured at different locations (e.g. P1 – P9)? These data would help the readers to better understand the spatial and temporal distribution of indoor constituents in this campaign.

Line 85, "While the gas-phase chemistry model and multiphase kinetic model treat comprehensive and detailed chemistry, it is computationally too expensive and unfeasible to treat all of these gas and multiphase reactions in the CFD. To circumvent this hurdle, we constrained the CFD with key inputs from the detailed models: the INDCM provided production rates and reactivity of OH radicals, while the multiphase kinetic model provided HOCl, ClNO₂, NCl₃, and NH₃ concentrations right above the bleach surface over time as controlled by aqueous reactions in the bleach." I agree with authors' arguments, but would like to ask if the authors have investigated the spatial and temporal profiles of VOCs in their measurements and CFD simulations. If yes, what would be the spatial and temporal distribution of VOCs? How would the data compare with those obtained from CFD simulations if available?

Line 103, "Model simulations reveal that the observed enhancement of OH radicals during the bleach cleaning event can be mainly explained by a cascade of reactions initiated via Cl₂ photolysis:" Could the authors comment to what extent the enhancement of OH radicals can be explained by the Cl₂ photochemical chemistry?

Line 112, "Horizontal and vertical distributions in Fig. 2 show that high concentrations of OH radicals are confined only to the solar radiation zone where they are generated via photolysis, while their concentration is low in the dark zone due to depletion through loss reactions." Could the authors comment how to verify the horizontal and vertical distribution obtained from CFD simulations with the measurement data?

Line 131, "Then, spatial scales or the average distance traveled can be estimated by considering a typical indoor air flow velocity of 0.03 m s⁻¹, corresponding to an air exchange rate of 0.5 h⁻¹ 21. The results of this analysis are depicted in Fig. 3, in which three distinct scales emerged:" Could the authors briefly comment how the horizontal and vertical distributions could be affect by the air exchange rate from the aspects of gas-phase species measurement and CFD simulation in this study?

Line 171, "Fig. 3 also includes spatial and temporal variations of particulate matter (PM) with

different particle diameters of 3 nm, 10 nm, 1 μm , 10 μm , and 100 μm , which determine the particle deposition velocity and residence time in indoor environments^{14,27} (see Table S2).” Do the PM allow to grow or shrink by condensation and evaporation processes (e.g. VOCs) in the simulations?

Reviewer #2 (Remarks to the Author):

This is an excellent manuscript that concisely summarizes detailed spatio-temporally resolved modeling of indoor air chemical composition during a bleach-cleaning event inside a house. The results match the experimental observations in trend lines and magnitude, which is quite exciting. This reviewer is not familiar with the journal, but there appears to be a need for a summary/conclusions section, and perhaps a paragraph laying out where this type of work is going in the future. Given the use of their model, the authors may wish to turn off certain parts of it to evaluate which contribution is the most and the least important in recapitulating the experimental data. Heterogeneous processes are probably the most unknown in this field, so turning off that aspect of the model probably has a large impact on how well it matches the experiments. Citing the two recent review in Chem, 6(12), 3203-18 (202) and outlook in Cell Reports Physical Science, 1, 11, 100256 (2020), as well as the very recent paper on multi-layer chemistry in Indoor Air (<https://doi.org/10.1111/ina.12854>) by some of the authors is probably appropriate in that context. This special role of surface chemistry that's just now beginning to be understood is also important in the sars-cov motivated context of the present manuscript - a recent paper in Chem, 6 (9), 2135-46 (2020) makes this point very nicely. These are suggestions for the conclusions/outlook section that will further improve the already high quality of the manuscript. The graphics are excellent. This reviewer would be happy to review a revised manuscript.

Reviewer #3 (Remarks to the Author):

This is a very interesting manuscript and clearly an immense amount of work has gone into the experimental and modeling efforts.

My primary comment is that I am confused why there is no Methods section and why all of the methodology is presented in the Supplemental Information. I found it challenging to put the results into context without seeing the methodology.

I believe adding a Methods section to the main paper and providing additional context to orient readers would improve the accessibility of this manuscript.

Specific comments:

In Figure 1c, what do the error bars represent?

In Supplementary Table 2, can sources be provided for the model inputs?

In Section S.5, does the reference formatting change? I am unclear on what (8) and (7) refer to.

In Supplementary Figure 2, can you provide a theory for why the measured vs modeled data do not match as well as for the species presented in the main manuscript?

Without the Methods in the main doc and summarizing methods from previous publications in the SI, I found it challenging to keep straight where CFD inputs were coming from - which were measured in this or previous experiments or were the outputs of other models or assumptions. For example, did you measure the AER during the experiment or use a reasonable assumed AER? A little more context would help the reader.

I know it is easy to become so close to your work that it is hard to step back and view your manuscript as a novice, but I think the manuscript would benefit from this. There are many small things, such as the red line in Figure 1b isn't explained until Figure 2, but leaves the reader thinking they are missing something in Figure 1.

Please see our point-by-point response as below (comments in black, response in blue).

Reviewer #1 (Remarks to the Author):

In this work, the authors demonstrated that spatial and temporal scales of indoor constituents including gas-phase oxidants, gas-phase species and PM can be modulated by rates of chemical reactions, surface interactions and building ventilation during the bleaching experiment in the HOMEChem campaign in a combination of measurements and CFD simulations. The findings of this work provides greater insights for better assessing the human exposure to gas-phase species and PM in indoor environments. The paper is very well written and concise. I support the publication of this work with minor comments.

We thank Reviewer 1 for the review and very positive evaluation of our manuscript.

Comments

Line 71, “Most measurements were conducted at one location (P2) in the kitchen, while OH was measured in the sunlit zone next to the window at P7 (see Fig. 1b).” Why P2 in the kitchen was chosen for the most of the measurements? What were the results measured at different locations (e.g. P1 – P9)? These data would help the readers to better understand the spatial and temporal distribution of indoor constituents in this campaign.

Most measurements were conducted in the kitchen at P2, as emissions were expected to be high and also due to sampling logistic. OH and HONO were measured at P7 right next to window as their concentrations were expected to be significant in the sunlit zone. Other points represent calculation points in the CFD model.

Line 85, “While the gas-phase chemistry model and multiphase kinetic model treat comprehensive and detailed chemistry, it is computationally too expensive and unfeasible to treat all of these gas and multiphase reactions in the CFD. To circumvent this hurdle, we constrained the CFD with key inputs from the detailed models: the INDCM provided production rates and reactivity of OH radicals, while the multiphase kinetic model provided HOCl, ClNO₂, NCl₃, and NH₃ concentrations right above the bleach surface over time as controlled by aqueous reactions in the bleach.” I agree with authors’ arguments, but would like to ask if the authors have investigated the spatial and temporal profiles of VOCs in their measurements and CFD simulations. If yes, what would be the spatial and temporal distribution of VOCs? How would the data compare with those obtained from CFD simulations if available?

We did not resolve VOCs in the CFD model, so unfortunately their spatial and temporal profiles are not available. Based on the analysis of temporal and spatial scales of isoprene in Fig. 3, we expect that VOCs would be relatively homogeneous. This aspect should be further investigated by measurements and modeling in future studies.

Line 103, “Model simulations reveal that the observed enhancement of OH radicals during the bleach cleaning event can be mainly explained by a cascade of reactions initiated via Cl₂

photolysis:” Could the authors comment to what extent the enhancement of OH radicals can be explained by the Cl₂ photochemical chemistry?

The model simulations by INDCM indicate that a cascade of reactions initiated via Cl₂ photolysis (the formed Cl radicals react with VOCs to generate peroxy and alkoxy radicals, which propagate to OH) represents a dominant OH formation pathway, which is by a factor of 10 or more important than HOCl photolysis. This point is clarified in the revised manuscript as below:

L103-108: “Model simulations reveal that the observed enhancement of OH radicals during the bleach cleaning event can be mainly explained by a cascade of reactions initiated via Cl₂ photolysis: the formed Cl radicals react with volatile organic compounds (VOCs) to generate peroxy and alkoxy radicals, which propagate to HO₂ and then OH through reactions involving NO. Gas-phase model simulations indicate that this process accounts for >90% of OH production, while OH radicals can also be generated via photolysis of HOCl and HONO⁵.”

Line 112, “Horizontal and vertical distributions in Fig. 2 show that high concentrations of OH radicals are confined only to the solar radiation zone where they are generated via photolysis, while their concentration is low in the dark zone due to depletion through loss reactions.” Could the authors comment how to verify the horizontal and vertical distribution obtained from CFD simulations with the measurement data?

As OH was measured only at one location at P7, it is challenging to directly validate horizontal and vertical distributions. However, when P7 was in dark conditions on this day as well during other bleach cleaning events, OH concentrations were near the detection limit of the instrument (approximately $1 \times 10^6 \text{ cm}^{-3}$). Elevated concentrations of OH during bleach cleaning events were only observed when P7 was illuminated. These observations serve as an indirect validation of strong concentration gradients between dark and sunlit zones. We added this explanation in the revised SI:

“As OH was measured only at P7, it is challenging to directly validate horizontal and vertical distributions. However, when P7 was under dark conditions during other bleach cleaning events, OH concentrations were near the detection limit of the instrument (approximately $1 \times 10^6 \text{ cm}^{-3}$). Elevated concentrations of OH during bleach cleaning events were only observed when P7 was illuminated. These observations serve as an indirect validation of strong concentration gradients between dark and sunlit zones.”

Line 131, “Then, spatial scales or the average distance traveled can be estimated by considering a typical indoor air flow velocity of 0.03 m s⁻¹, corresponding to an air exchange rate of 0.5 h⁻¹ 21. The results of this analysis are depicted in Fig. 3, in which three distinct scales emerged:” Could the authors briefly comment how the horizontal and vertical distributions could be affected by the air exchange rate from the aspects of gas-phase species measurement and CFD simulation in this study?

The analysis presented in Fig. 3 is consistent with gas-phase measurements and CFD results presented in Figs. 1 and 2, as discussed in the manuscript. The air exchange rate indeed affects the temporal and spatial scales significantly. At higher air exchange rates that are often deployed in industrial buildings with mechanical ventilation, the temporal and spatial scales of moderately long-lived and long-lived species would both decrease as the species are transported to the ambient atmosphere at a faster rate (see Fig. S3 for such analysis with an air exchange rate of 5 h⁻¹).

Line 171, “Fig. 3 also includes spatial and temporal variations of particulate matter (PM) with different particle diameters of 3 nm, 10 nm, 1 μm, 10 μm, and 100 μm, which determine the particle deposition velocity and residence time in indoor environments^{14,27} (see Table S2).” Do the PM allow to grow or shrink by condensation and evaporation processes (e.g. VOCs) in the simulations?

While PM is analyzed for spatial and temporal scales presented in Fig. 3, we did not treat PM and their processes (e.g., condensation/evaporation) in CFD simulations. We clarify this point in the revised SI:

“It was assumed that aerosol particles, where HOCl uptake occurred on the surface, were distributed uniformly throughout the room, while particles were not explicitly resolved in CFD.”

Reviewer #2 (Remarks to the Author):

This is an excellent manuscript that concisely summarizes detailed spatio-temporally resolved modeling of indoor air chemical composition during a bleach-cleaning event inside a house. The results match the experimental observations in trend lines and magnitude, which is quite exciting. This reviewer is not familiar with the journal, but there appears to be a need for a summary/conclusions section, and perhaps a paragraph laying out where this type of work is going in the future. Given the use of their model, the authors may wish to turn off certain parts of it to evaluate which contribution is the most and the least important in recapitulating the experimental data. Heterogeneous processes are probably the most unknown in this field, so turning off that aspect of the model probably has a large impact on how well it matches the experiments. Citing the two recent review in Chem, 6(12), 3203-18 (202) and outlook in Cell Reports Physical Science, 1, 11, 100256 (2020), as well as the very recent paper on multi-layer chemistry in Indoor Air (<https://doi.org/10.1111/ina.12854>) by some of the authors is probably appropriate in that context. This special role of surface chemistry that's just now beginning to be understood is also important in the sars-cov motivated context of the present manuscript - a recent paper in Chem, 6 (9), 2135-46 (2020) makes this point very nicely. These are suggestions for the conclusions/outlook section that will further improve the already high quality of the manuscript. The graphics are excellent. This reviewer would be happy to review a revised manuscript.

We thank Reviewer 2 for the review and very positive evaluation of our manuscript. We agree that the heterogeneous processes on indoor surfaces is critical. In Mattila et al., ES&T, 54, 1730

(2020), in which we modeled the same data set of bleach cleaning, we demonstrated that multiphase chemistry is crucial in generating chlorinated and nitrogenated compounds (e.g., observations could not be reproduced when heterogeneous processes were switched off). We clarify this point in the revised SI:

“Note that observations of chlorinated and nitrogenated compounds could not be reproduced by the model without multiphase chemical processes, as demonstrated in our previous study¹.”

We add a summary/conclusion section in the revised manuscript, discussing the importance of indoor surface processes by citing the suggested references as below:

“In summary, we demonstrate that heterogeneous distributions of indoor air pollutants can exist for short-lived and moderately long-lived compounds, in contrast to the traditional assumption of homogeneous mixing. The spatial and temporal scales are controlled by gas-phase and multiphase reactions, deposition as well as indoor air flow and outdoor-indoor air exchange. Among these factors, surface interactions may be least characterized and quantified, despite their importance becoming increasingly clear^{24,31}. Different surface and environmental conditions including temperature, humidity, light, and surface pH would be critical for heterogeneous reactions at indoor surfaces^{24,31} as well as surface stability of SARS-COV2.³² In addition, the presence of organic films on indoor surfaces can impact thermodynamics and kinetics of SVOC partitioning^{33,34}. Further elucidation of these aspects will improve assessments on indoor air quality, human exposure to indoor pollutants and indoor-outdoor transport of chemical compounds.”

Reviewer #3 (Remarks to the Author):

This is a very interesting manuscript and clearly an immense amount of work has gone into the experimental and modeling efforts.

My primary comment is that I am confused why there is no Methods section and why all of the methodology is presented in the Supplemental Information. I found it challenging to put the results into context without seeing the methodology. I believe adding a Methods section to the main paper and providing additional context to orient readers would improve the accessibility of this manuscript.

We thank Reviewer 3 for the review and very positive evaluation of our manuscript. Following your and editor’s suggestions, we add a method section in the main text.

Specific comments:

In Figure 1c, what do the error bars represent?

The error bars represent the 1σ standard error of the precision of the OH measurements and are separate from the calibration accuracy ($\pm 18\%$, 1σ). We clarify this in the revised caption.

In Supplementary Table 2, can sources be provided for the model inputs?

References are provided in Sect. 5.1. They have now also been added to the caption of Table S2.

In Section S.5, does the reference formatting change? I am unclear on what (8) and (7) refer to.

Thanks for catching this formatting error, which is fixed in the revised SI.

In Supplementary Figure 2, can you provide a theory for why the measured vs modeled data do not match as well as for the species presented in the main manuscript?

As pointed out, there are some discrepancies between measurements and models for Cl_2 and ClNO_2 . The difference between measurements and kinetic model for Cl_2 is most likely due to the assumption of homogeneous mixing in kinetic modeling. In the matter of fact, CFD modeling resolving spatial distributions agree better with measurements. For ClNO_2 , the measurement showed initial peaks within 10 minutes, which was likely due to primary emissions as impurity in bleach solutions as observed in a previous study (Wong et al., 2017). However, this process was not treated in our models as we were unable to assess this source in our models due to lack of experimental constraints (e.g., Mattila et al., 2020). This point is clarified in the revised SI:

“A previous study has suggested primary emissions of Cl_2 and ClNO_2 from the bleach due to solution impurities.⁵ This possibility was not considered in the model due to a lack of experimental constraints,⁶ which may be one of the reasons for the difference between measurements and modeling (Fig. S2).”

Without the Methods in the main doc and summarizing methods from previous publications in the SI, I found it challenging to keep straight where CFD inputs were coming from - which were measured in this or previous experiments or were the outputs of other models or assumptions. For example, did you measure the AER during the experiment or use a reasonable assumed AER? A little more context would help the reader.

The CFD inputs for simulating indoor airflow were from the previous experiments while the concentrations of HOCl , ClNO_2 , chloramines, and NH_3 directly above the bleach surface were inputted from the kinetic model. House ventilation and household state were both controlled well and monitored during the HOMEChem campaign, as detailed in Farmer et al., Environ. Sci. Processes Impacts, 21, 1280, 2019. We added this info in the revised SI:

“The model also simulated outdoor air infiltration into the house at a rate of 0.7 h^{-1} and indoor air recirculation through the central air handling unit at an air mixing rate of 8 h^{-1} , as characterized during the HOMEChem campaign⁷.”

I know it is easy to become so close to your work that it is hard to step back and view your

manuscript as a novice, but I think the manuscript would benefit from this. There are many small things, such as the red line in Figure 1b isn't explained until Figure 2, but leaves the reader thinking they are missing something in Figure 1.

Thanks for pointing this out and we add this explanation in the caption of Figure 1:

“The vertical red line represents the cross section used for the vertical maps presented in Fig. 2.”

References.

Mattila, J. M., Lakey, P. S. J., Shiraiwa, M., Wang, C., Abbatt, J. P. D., Arata, C., Goldstein, A. H., Ampollini, L., Katz, E. F., DeCarlo, P. F., Zhou, S., Kahan, T. F., Cardoso-Saldaña, F. J., Ruiz, L. H., Abeleira, A., Boedicker, E. K., Vance, M. E. and Farmer, D. K.: Multiphase Chemistry Controls Inorganic Chlorinated and Nitrogenated Compounds in Indoor Air during Bleach Cleaning, *Environ. Sci. Technol.*, 54, 1730-1739, 2020

Wong, J. P. S., Carslaw, N., Zhao, R., Zhou, S. and Abbatt, J. P. D.: Observations and impacts of bleach washing on indoor chlorine chemistry, *Indoor Air*, 27, 1082-1090, 2017

REVIEWERS' COMMENTS:

Reviewer #1 (Remarks to the Author):

The authors have fully addressed the comments raised by the reviewers. I support the publication of this revision.

Reviewer #3 recommends publication as is.